# Vitamin D Insufficiency Reduces Grip Strength, Grip Endurance and Increases Frailty in Aged C57Bl/6J Mice

**DOI:** 10.3390/nu12103005

**Published:** 2020-09-30

**Authors:** Kenneth Ladd Seldeen, Reem Nagi Berman, Manhui Pang, Ginger Lasky, Carleara Weiss, Brian Alexander MacDonald, Ramkumar Thiyagarajan, Yonas Redae, Bruce Robert Troen

**Affiliations:** 1Division of Geriatrics and Palliative Medicine, Jacobs School of Medicine and Biomedical Sciences, University at Buffalo and Research Service, Buffalo, NY 14203, USA; seldeen@buffalo.edu (K.L.S.); reemberm@buffalo.edu (R.N.B.); mpang23@gmail.com (M.P.); ginger.lasky@astrazeneca.com (G.L.); carleara@buffalo.edu (C.W.); brianmacdonald1@gmail.com (B.A.M.); rthiyaga@buffalo.edu (R.T.); yonasred@buffalo.edu (Y.R.); 2Veterans Affairs Western New York Healthcare System, Buffalo, NY 14215, USA

**Keywords:** cholecalciferol, 25-OH vitamin D, parathyroid hormone, aging, physical performance, frailty, body composition, bone density

## Abstract

Low 25-OH serum vitamin D (VitD) is pervasive in older adults and linked to functional decline and progression of frailty. We have previously shown that chronic VitD insufficiency in “middle-aged” mice results in impaired anaerobic exercise capacity, decreased lean mass, and increased adiposity. Here, we examine if VitD insufficiency results in similar deficits and greater frailty progression in old-aged (24 to 28 months of age) mice. Similar to what we report in younger mice, older mice exhibit a rapid and sustained response in serum 25-OH VitD levels to differential supplementation, including insufficient (125 IU/kg chow), sufficient (1000 IU/kg chow), and hypersufficient (8000 IU/kg chow) groups. During the 4-month time course, mice were assessed for body composition (DEXA), physical performance, and frailty using a Fried physical phenotype-based assessment tool. The 125 IU mice exhibited worse grip strength (*p* = 0.002) and inverted grip hang time (*p* = 0.003) at endpoint and the 8000 IU mice transiently displayed greater rotarod performance after 3 months (*p* = 0.012), yet other aspects including treadmill performance and gait speed were unaffected. However, 125 and 1000 IU mice exhibited greater frailty compared to baseline (*p* = 0.001 and *p* = 0.038, respectively), whereas 8000 IU mice did not (*p* = 0.341). These data indicate targeting higher serum 25-OH vitamin D levels may attenuate frailty progression during aging.

## 1. Introduction

Frailty is a condition of greater susceptibility to adverse outcomes, including disability, loss of independence, and mortality. Additionally, frailty increases progressively with age, existing in 10% of individuals 65 and over and as much as 50% of those over the age of 85 [1,2]. Concurrently, vitamin D insufficiency (25-OH vitamin D < 30 ng/mL) is a pervasive public health challenge that affects as many as 70% of the world population [3], and may contribute to frailty [4]. Older adult populations are of particular concern for low serum vitamin D due to decreased sun exposure and a natural age-associated decline in the generation of cholecalciferol (vitamin D3) from sunlight in skin tissues [5,6]. Therefore, addressing the relationships between frailty and serum 25-OH vitamin D may allow for greater understanding of the mechanisms underlying the maintenance of functional capacity and facilitate the promotion of healthier aging.

Additionally, age-related musculoskeletal declines such as osteoporosis and sarcopenia are directly related to aging and can catalyze frailty in older populations. Of particular interest are the contributions of sarcopenia, the loss of muscle mass and function over time, to the frailty phenotype [7]—especially in light of the important role vitamin D appears to play in the development of sarcopenia [8]. Furthermore, we have previously published that chronic vitamin D insufficiency in middle-aged mice induces physical performance decline in grip endurance and anaerobic performance [9]. In that study, young 6-month old mice consumed either sufficient (1000 IU) or insufficient (125 IU) vitamin D3/kg chow while assessments tracking strength, aerobic endurance, and exploratory behavior were done in a 12-month time frame [9]. Others have also identified a potential role for vitamin D impacts in mice, particularly through the use of vitamin D receptor knockout models that have identified worse rotarod coordination and swim endurance [10,11,12]. To our knowledge, the impacts of serum vitamin D levels on physical performance in old aged mice have not been investigated.

Additionally, individuals with low serum levels of vitamin D are significantly at a higher risk of frailty [13,14]. Yet, the relationships between frailty and vitamin D have heretofore not been studied in a mouse model that can reduce genetic and lifestyle contributions via the use of genetically identical mice and maintaining mice with similar housing, lighting, and chow compositions. Additionally, mouse frailty assessment tools are emerging and aid in the translatability of preclinical studies [15,16]. We have previously used a Fried Frailty Phenotype mouse assessment tool that mimics human-based tools for assessing frailty in mice models to investigate the impacts of exercise in mice [17,18,19]. Here, the aim of our investigation is to elucidate the impacts of vitamin D insufficiency and the potential therapeutic benefits of vitamin D hypersufficiency on frailty, physical performance, and body composition for the first time in old aged mice. Our study identifies that vitamin D status affects specific aspects of physical performance and that only hypersufficient levels of vitamin D attenuate the progression of frailty.

## 2. Materials and Methods

### 2.1. Animals

All studies and experimental protocols were approved by and in compliance with guidelines of the University at Buffalo (Protocol#: MED16083N) and VA Western New York (Protocol#: 532537) Animal Care and Use Committees. Forty-two C57BL/6J male mice were acquired at 22 months of age from the National Institutes of Aging mouse colony. At 24-months of age, mice were assorted into groups (125 IU, 1000 IU, 8000 IU) based upon body weight and switched from facility chow to AIN93G-based chow (Dyets Inc., Bethlehem, PA, USA) containing either 125, 1000, or 8000 IU vitamin D3 per kg chow, respectively. After the 4-month experimental time frame, final group numbers following attrition due to natural causes were 12 for the 125 IU, 8 for the 1000 IU, and 11 for the 8000 IU groups. Specialized chow and water were provided ad libitum and mice were housed in large animal cages containing 3 or 4 mice per cage, with lighting provided on a 12-h on/off cycle. In addition, filters were installed in all mouse holding and assessment rooms to block UV contributions from fluorescent lighting. Body weight was measured weekly.

### 2.2. Serum Analyses

At each timepoint, blood was acquired via the submandibular vein, placed on ice for 1-h, and then, centrifuged at 16,000× *g* to collect serum. 25-OH vitamin D was assessed at each timepoint using ELISA (ImmunoDiagnostic Systems, Inc., Scottsdale, AZ, USA). At baseline and endpoint, intact parathyroid hormone (PTH) was also determined using ELISA (MyBioSource, San Diego, CA, USA).

### 2.3. Dual X-ray Absorptiometry

To measure bone mineral density, body fat %, and lean mass, mice were anesthetized using ketamine and xylazine and then, scanned using a Lunar PIXImus II (Inside Outside Sales, LLC., Fitchburg, WI, USA). DEXA scans were performed at baseline and endpoint. Analysis was performed using system software allowing determination of body lean and fat mass as well as bone mineral density.

### 2.4. Physical Performance Assessments

Mice were acclimated to physical performance equipment one month prior to baseline. Baseline and endpoint assessments were carried out for each behavioral assessment by the same investigator, who was blind to the treatment of the mice. All experiments were carried out during lighted hours and at the same time of day between baseline and endpoint. Assessments were conducted monthly until endpoint.

#### 2.4.1. Rotarod Assessment 

Rotarod assessment was assessed using a 5-lane Rota-Rod instrument for mice (MedAssociates Inc., Fairfax, VT, USA). Mice were initially acclimated to the device with three trials at a speed of 2 to 20 RPM over 5 min one month prior to baseline. For each assessment, mice were timed as the average of the best 2 of 3 trials, in which the device accelerated from 4 to 40 RPM over 5 min. Trials were ended if the mouse fell from the cylinder or held the cylinder for 2 complete revolutions.

#### 2.4.2. Gait Speed Assessment 

Gait speed assessment was assessed using a device constructed according to methodology described in [20]. Briefly, two weeks before baseline, mice were acclimated to a darkened safe house at the end of an approximately 1-m narrow track and the whole device. Mice were placed at the start of the track and timed to return to the safe house, and the best 2 of 3 trials were averaged to generate the final time.

#### 2.4.3. Open Field Activity 

Open field activity was assessed using an open field arena with infrared photo beam arrays (MedAssociates Inc.). Each arena was 40 × 40 × 35 cm and mice were placed in the chamber for a total of 30 min. System software was used to determine the total numbers of quadrant crossings and rearings.

#### 2.4.4. Grip Strength 

Grip strength was assessed using a grip force meter (Columbus Instruments, Columbus, OH, USA). For each trial, mice were placed on the device and pulled parallel to the ground until loss of grip. At each timepoint, mice were given 5 trials with 10 s of rest between trials. The best 3 of 5 trials were averaged to generate a final score per mouse.

#### 2.4.5. Inverted Grip Endurance 

Inverted grip endurance was assessed using a constructed device that included a 30 × 30 cm wire grid with 1.4 cm cross-hatching that rested atop a 40 cm open ended box. For each timepoint, mice were scored on the better of two trials whereby the mouse was flipped over on the grid and timed until loss of grip and fall or reaching a maximum gripped duration of 300 s. Additionally, mice were given three attempts to attain at least 15 s.

#### 2.4.6. Treadmill Assessment 

Treadmill assessment was performed on a mouse treadmill (Columbus Instruments) set with no inclination (flat—0°), as we described previously [21]. Briefly, at each timepoint, an endurance assessment was performed whereby the treadmill belt accelerated from 5 to 35 m/min over 60 min and the time on the belt was recorded at exhaustion for each mouse, defined as 10 visits to a shock pad or 20 total shocks (shocks at 54 V, 0.72 mA). For uphill sprint interval assessment, the treadmill was set to a 25° inclination. Belt speed started at 5 m/min for 30 s and with incrementing intervals of 15 s at a test speed followed by 20 s active recovery at 5 m/min. Test intervals started at 10 m/min and increased to 1 m/min each interval. Exhaustion was defined as 5 visits to the shock pad or 10 total shocks, and the final score given as the speed of the last completed interval. For maximal speed assessment, the treadmill was set with no inclination. The timing and exhaustion parameters were the same as the uphill sprint, except speeds incremented 3 m/min after each interval. For all assessments, mice were given a single trial per timepoint.

### 2.5. Frailty Determination

Frailty assessment was performed as described previously [19]. Briefly, the frailty tool is patterned based on the Fried et al. physical frailty assessment for humans [17], and likewise, this tool for mice includes 5 parameters, unexpected weight loss (>5% weight loss one week prior to baseline and prior to endpoint), and a performance score below a cutoff of 1.5 standard deviations of the baseline group mean of the 50% of mice closest to the mean for grip strength, treadmill endurance, gait speed, and open field activity (quadrant crossings). For this method, mice received 1 point for each parameter below the cut-point, and mice with ≥3 such parameters were considered frail.

### 2.6. Statistics

Descriptive data were reported with means and standard deviation. A repeated measure analysis of variance (ANOVA) with Bonferroni corrections and Tukey’s Multiple Comparisons was used to estimate the effects of time (timepoints of treatment 0, 4, 8, 12, and 16), vitamin D treatment (groups 125, 1000, and 8000 IU), as well as time × group effects. Greenhouse–Geisser and Huynh–Feldt corrections were used when the sphericity assumption was violated. A paired *t*-test was used to compare within-group changes between specific timepoints. Statistical analysis was conducted with SPSS version 27 (IBM Corp., Armonk, NY, USA). Graphs were created with GraphPad Prism version 8.0.0 for Windows (GraphPad Software San Diego, CA, USA). Data are presented with the mean ± standard deviation. Results indicated with * were considered significant at *p* = 0.05.

## 3. Results

### 3.1. Changes in Vitamin D Supplementation Rapidly and Consistently Altered 25-OH Vitamin D Serum Levels in Aged Mice, Yet Did Not Affect Serum PTH or Bone Density

To elucidate the response to acute vitamin D insufficiency or hypersufficiency at advanced ages, we provided 24-month-old male C57Bl6 mice with either 125 (125 IU, *n* = 12), 1000 (1000 IU, *n* = 8), or 8000 (8000 IU, *n* = 11) IU of vitamin D3/kg chow and measured serum 25-OH vitamin D levels monthly (Figure 1A). Serum 25-OH vitamin D levels in the 125 IU mice rapidly declined after 1 month (125 IU: baseline: 34.8 ± 8.9 ng/mL versus 4 weeks: 14.2 ± 1.8 ng/mL, *p* < 0.0001), and subsequently stabilized between 11 and 14 ng/mL for the duration of the experiment, which was significantly different than the 1000 IU group (*p* < 0.001). Serum 25-OH vitamin D levels in the 8000 IU group likewise increased rapidly to a new equilibrium after 1 month (8000 IU: baseline: 42.5 ± 9.8 ng/ml versus 4 weeks: 62.9 ± 23.1 ng/mL, *p* < 0.0061), and was significantly higher than in the 1000 IU group (*p* = 0.039).

The parathyroid hormone (PTH) vitamin D axis foundationally mediates bone metabolism and has been implicated in physical performance and muscle function [22]. We therefore set out to identify changes in serum-intact PTH in response to altered vitamin D supplementation in aged mice using ELISA (Figure 1B). However, our data revealed no statistically significant differences between groups for the time period of this experiment. We further investigated impacts upon bone mineral density (BMD) using dual X-ray absorptiometry (Figure 1C and Appendix A). Although no differences were observed between groups, compared to baseline, a trend was observed for increased bone density in the 8000 IU mice (50.9 ± 1.8 versus 51.7 ± 2.0 mg/cm^2^, *p* = 0.121).

### 3.2. Serum Levels of 25-OH Vitamin D Did Not Associate with Differences in Body Weight between Groups, but Vitamin D Insufficient Exhibited Increased Body Fat Percentage

Vitamin D has previously been identified as a contributing factor during weight loss [23,24]. We therefore set out to determine if body weight and body composition were altered by changes in vitamin D status in aged mice. We first undertook weekly measurements of body weight following changes in vitamin D supplementation, yet did not find any statistically significant differences between or within supplementation groups (Figure 2A). We next examined changes in body fat and lean mass, and identified a statistically significant decline in body fat in the 1000 IU mice (21.0 ± 3.3% versus 17.7 ± 3.3%, *p* = 0.007), a trend of decline in the 8000 IU mice (21.6 ± 4.1% versus 19.7 ± 5.0%, *p* = 0.09), and no difference in the 125 IU mice (21.0 ± 4.0% versus 20.6 ± 5.4%, *p* = 0.78, Figure 2B). No differences in lean mass were observed between or within groups, with 1000 IU mice nearly reaching significance (*p* = 0.06, Figure 2C).

### 3.3. Vitamin D Insufficient Mice Displayed Lower Grip Strength and Inverted Grip Endurance without Effect on Aerobic or Anaerobic Treadmill Performance

We previously identified physical performance deficits in 6-month-old C57BL/6 mice made vitamin D insufficient for 12 months [9]. Here, we investigated the impacts of 4 months of vitamin D insufficiency or hypersufficiency in 24-month-old mice. We found that the 125 IU mice exhibited declines in grip strength (baseline: 1.91 ± 0.21 N versus endpoint: 1.61 ± 0.22 N, *p* = 0.002, Figure 3A). In contrast, no differences were observed in the 1000 IU mice (baseline: 1.68 ± 0.21 N versus endpoint: 1.62 ± 0.18 N, *p* = 0.290), while a trend was found in the 8000 IU mice (baseline: 1.82 ± 0.17 N versus endpoint: 1.70 ± 0.22 N, *p* = 0.172). Additionally, the 125 IU mice exhibited worse inverted grip endurance after 12 weeks (baseline: 3.4 ± 1.3 min versus 12 weeks: 2.7 ± 1.2 min, *p* = 0.003) and continuing at endpoint (endpoint: 2.4 ± 1.4 min, *p* = 0.002, Figure 3B). We also observed a decline in the 1000 IU mice at endpoint only (baseline: 3.5 ± 1.2 min versus endpoint: 2.3 ± 0.5 min, *p* = 0.026), while the 8000 IU mice did not show declines at any point of this experiment (Figure 3B). However, we did not identify any vitamin D supplementation impacts on the three treadmill assessments with increasing anaerobic demand, including slow acceleration on the flat treadmill (Figure 3C), uphill incrementing speeds (Figure 3D), and flat with rapidly incrementing speeds (Figure 3E). Of note, only maximal flat speed showed an age-associated decline across all groups (*p* < 0.001).

### 3.4. Vitamin D Hypersufficiency Transiently Improves Balance and Coordination, However Gait Speed and Activity Levels Were Not Affected by Vitamin D Status

We further investigated the impacts of differential supplementation on aspects of physical performance including balance, gait speed, and activity levels that are highly relevant to older patients. In particular, lower 25-OH vitamin D serum levels have been associated with an increased risk of falling [4,25]. To investigate balance and coordination in mice, we assessed rotarod performance in differentially supplemented mice monthly (Figure 4A). Although we did not identify significant differences for any group at endpoint, 8000 IU mice were trending towards improved rotarod performance after 8 weeks and were significantly improved at 12 weeks, then returning to baseline levels at endpoint (8000 IU baseline: 145.5 ± 26.9 s versus 8 weeks: 170.5 ± 54.8 s, *p* = 0.141; 12 weeks: 184.6 ± 55.2 s, *p* = 0.012; endpoint: 158.3 ± 66.9 s, *p* = 0.635). We next analyzed gait speed but did not find any response to altered supplementation (Figure 4B). All mice improved dramatically from baseline to the assessment at week 4; however, this is believed to be due to inadequate initial acclimation to the procedure. Additionally, we undertook monthly assessments of open field activity (Figure 4C,D), as we previously identified reduced open field rearing in middle-aged vitamin D insufficient mice [9]. However, here, we did not observe differences due to vitamin D supplementation at any timepoint.

### 3.5. Frailty Scores Increased in the Vitamin D Insufficient and Sufficient Mice, but Did Not in the Hypersufficient Vitamin D Mice

Human epidemiologic studies have identified a potential correlation between vitamin D status and frailty, yet identifying the direct impacts of serum 25-OH vitamin D levels on frailty status is difficult in clinical settings, potentially due to the need for longer term longitudinal studies. Here, we used our animal frailty assessment tool based on the Fried physical frailty phenotype scale [17,19]. For the five Fried frailty parameters of weight change, grip strength, endurance, activity, and gait speed, our mouse assessment used weight loss in a one-week time period before baseline and endpoint, grip meter, treadmill performance, open field activity, and the gait speed test, respectively. After 4 months of differential vitamin D supplementation, we found that both the 125 and the 1000 IU groups exhibited greater frailty (parameters below cutoff: 125 IU—baseline: 0.9 ± 0.5 to endpoint: 2.1 ± 1.5, *p* = 0.001; 1000 IU—baseline: 1.0 ± 1.2 to endpoint: 2.1 ± 1.4, *p* = 0.038, Figure 5). However, the 8000 IU mice showed similar frailty scores from baseline to endpoint (8000 IU—baseline: 1.4 ± 0.9 to endpoint: 1.6 ± 1.0, *p* = 0.341).

## 4. Discussion

Vitamin D insufficiency is a widespread condition that represents a potential risk factor for older adults. Here, we investigate the impacts of differential vitamin D supplementation in aged mice. We found that a hypersufficiency state prevented the increase in frailty observed in the other groups after 4 months. We have previously demonstrated the ability to investigate the impacts of vitamin D insufficiency on obesity and physical performance in young and middle-aged mice [9,26]. In this study, we further demonstrate that aged mice also exhibit similar responses to differential doses of vitamin D in chow. In particular, the 25-OH vitamin D serum levels in the 125 IU mice exhibited rapid declines by 1 month, which remained within a tight range for the duration of the experiment (Figure 1A). Additionally, we observed a rapid equilibration with increased 25-OH vitamin D serum levels in the 8000 IU mice after one month that was likewise sustained thereafter for the time course of the experiment. These changes are in line with the kinetics of altered serum 25-OH vitamin D levels in response to changes in supplementation that we and others have previously reported [9,26,27]. However, these findings sit in contrast to our previous study that did not observe a statistically different impact of increasing concentration in chow to 4000 IU/kg chow [26]. Here, our data indicate 8000 IU/kg chow, unlike 4000 IU/kg chow, allows for discernable differences in serum 25-OH vitamin D levels, despite the observed variability in response to high dose chow, which may be due to aging. This was also seen to some degree in our previously reported 4000 IU/kg supplemented younger mice [26]. Furthermore, although vitamin D status is canonically associated with bone health, we did not observe impacts on bone density due to vitamin D insufficiency, similar to our previous studies in younger mice [9,26]. Interestingly, a trend towards greater bone density was observed in the 8000 IU group (Figure 1C), and others have reported that in young mice, 20,000 IU/kg chow, but not 8000 IU chow, for 4 weeks, enhanced bone quality [28]. These observations raise the possibility that a greater dose or longer period of treatment may have been needed to induce bone impacts.

The role of vitamin D in maintaining optimal physical performance in older individuals has been supported previously by multiple epidemiologic studies, which include observations that low serum levels of 25-OH vitamin D were associated with poor grip strength in centenarians [29], grip strength was also affected by low vitamin D in a small study involving 130 hemodialysis patients [30], short physical performance battery scores were significantly lower in community dwelling 70–89-year-old individuals who exhibited serum 25-OH vitamin D below 20 ng/ml [31], and optimal physical performance was observed in individuals with 25-OH vitamin D > 40 ng/ml in a study of 2694 community dwelling older adults [32]. Furthermore, we previously reported that mice kept vitamin D insufficient from 6 to 18 months of age exhibited worse anaerobic treadmill performance, grid hang time, fewer rearings during open field testing, and gait disturbances [9]. However, to our surprise, in this study, we observed few differences between vitamin D treatment groups for these assessments and others. A possible explanation is that we initiated vitamin D insufficiency in aged mice that were vitamin D sufficient up until that point in contrast to establishing vitamin D insufficiency at younger ages. The declines in physical performance due to vitamin D insufficiency may take decades for humans or a year or more in mice to develop. Additionally, one possible and unexpected confounder is that our monthly assessments of physical performance may have inadvertently contributed a “training effect” that positively improved the physical function of the mice. This notion of a “training effect” is supported by significant declines in performance being observed in only 2 of our 8 assessments (speed sprint and open field activity monitoring, Figure 3E and Figure 4C,D, respectively), which is generally uncharacteristic for what we and others have reported in mice aging from 24 to 28 months of age [18,33]. Future studies may need to consider the potential for such contributions in the study design to minimize confounders when assessing mice.

However, our analysis did detect differences due to vitamin D supplementation for three measures—grip strength, grip grid endurance, and rotarod performance. As discussed above, several cross-sectional studies have identified links between low 25-OH vitamin D levels and poor grip strength in older adults [29,30,34]. Our findings here suggest vitamin D insufficiency-induced loss of grip strength occurred progressively by 12 weeks (Figure 3A). Furthermore, studies that seek to enhance grip strength in older adults via vitamin D supplementation have seen no benefit [35,36], which is consistent with our observations of lack of benefit in the 8000 IU mice. Together, these data speak to the possibility that cross-sectional studies identifying vitamin D impacts on grip strength in humans may be capturing the effects of decades-long insufficiency in these participants.

Additionally, consistent with the decline in grip strength, we also identified a deficit in inverted grid hang time in the 125 IU mice (Figure 3B), for which grip strength likely contributes greatly to performance in this assessment. However, a second contributing factor may be the vestibular system, and our data reveal that vitamin D status may influence balance and coordination as assessed using rotarod (Figure 4A). Here, we identified that the 8000 IU mice demonstrated improved rotarod latency after 12 weeks relative to baseline, which dropped off thereafter (Figure 4A). These data are consistent with those reported by Sakai et al. who demonstrated that a 1,25(OH)_2_ vitamin D3 analog improved locomotor performance in young mice [37]. Together, these animal studies support a possible relationship between vitamin D and balance and coordination that have been identified in human trials, including likelihood to fall in older adults (reviewed in [38,39]). We also note the transient benefit of the 8000 IU supplementation in rotarod performance, which may indicate ultimately no benefit to supplement at endpoint, yet translated to humans may represent a clinically relevant improvement of an extra 10 years of healthspan.

Multiple studies have identified relationships between serum vitamin D levels and frailty status. In particular, 25-OH vitamin D levels below 20 ng/mL were found to correlate with greater frailty in two separate studies [40,41], while a third found levels greater than 15 ng/mL to be associated with less frailty [42]. Interestingly, an analysis of nearly 10,000 participants in the German-based ESTHER cohort revealed cross-sectional relationships between vitamin D status and frailty, but did not identify longitudinal relationships [43]. However, a study of 6307 women from four centers in the United States revealed that baseline levels of serum 25-OH vitamin D < 20 ng/mL were modestly associated with a greater risk of frailty [44,45]. We and others have reviewed the use of animal frailty assessment tools to characterize the condition in preclinical mouse studies to aid translatability to humans [15,16], exploring potential longitudinal relationships between vitamin D and frailty. Here, we use a longitudinal model that allows for observation of previously vitamin D sufficient aged mice that are then made insufficient for 4 months. In this context, only serum levels are changed while lifestyle and genetic factors are tightly controlled. This, in turn, allows for a more focused examination of the physiological and functional impacts of serum vitamin D that are otherwise difficult to accomplish in a human clinical trial. Here, we identified that while the 1000 IU and the 125 IU mice exhibited a statistically significant increase in frailty (Figure 5), the 8000 IU mice did not. These data suggest that higher than recommended supplementation and/or sun exposure may be necessary to attenuate frailty progression in older adults, which would require longer term human clinical trials (>5 years) to validate. We further note that the design of this experiment should be considered when evaluating these data in that these mice were vitamin D sufficient up until 24 months of age, and thus, we are, in effect, examining the ability of an older organism to respond to an acute change in vitamin D status. We also note the mice in our study were all male and that potential sex effects should be considered when extrapolating findings from this study to females. This is important given the findings of Burt et al. that high doses of vitamin D supplementation were accompanied by progressive losses in total volumetric BMD that were greater in females than in males [46]. Further research will be needed to parse out scenarios that are more clinically relevant, such as when humans or mice are vitamin D insufficient for significant periods of time leading up to old age, particularly in female populations.

## 5. Conclusions

Vitamin D insufficiency is a pervasive condition for which human clinical studies implicate a potential role in functional decline and frailty. These data indicate that higher serum levels (~60 ng/mL) prevent the progression of frailty and loss of grip strength in aged mice from 24 to 28 months of age. Furthermore, vitamin D insufficient mice exhibited poor inverted grip endurance, while vitamin D hypersufficient mice transiently demonstrated improved rotarod performance. This implies that the “osteocentric” serum 25-OH vitamin D goal of >30 ng/mL may be adequate for bone health, but may be suboptimal for other tissues such as muscle and the vestibulocochlear system, whereby vitamin D may mediate balance and coordination in aged mice. However, neither acute vitamin D insufficiency nor hypersufficiency affected other aspects of physical performance including aerobic and anaerobic treadmill endurance, gait speed, and open field activity. Future work is needed to identify the role of vitamin D in mediating physical function and frailty status across genders and during aging leading to old age.

## Figures and Tables

**Figure 1 nutrients-12-03005-f001:**
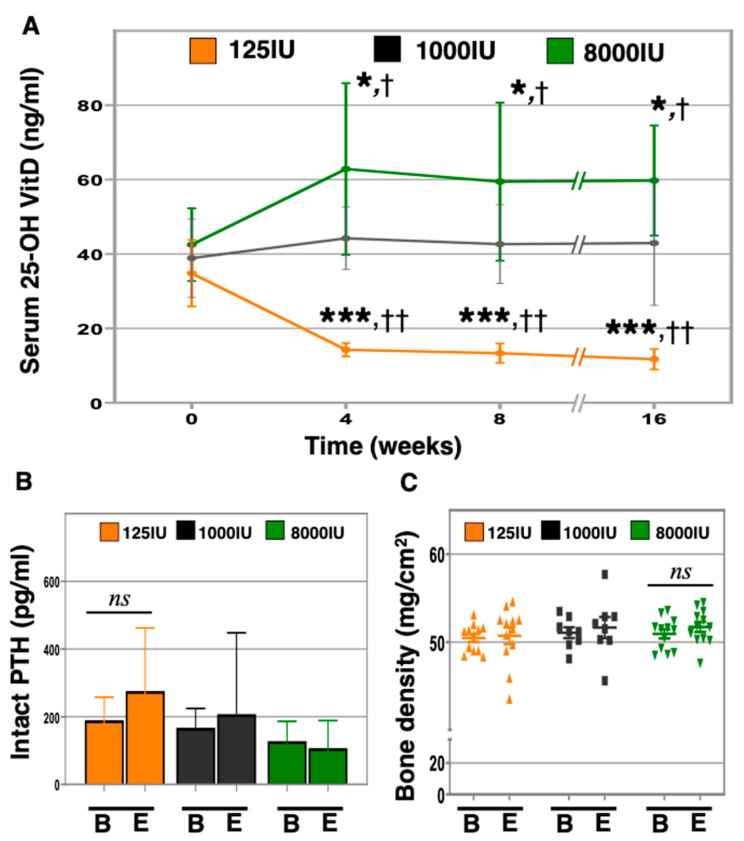
Impacts of altering 25-OH vitamin D (VitD) on intact parathyroid hormone (PTH) and bone density in aged mice. Twenty-four-month-old C57BL/6J male mice were given 125 (125 IU—orange), 1000 (1000 IU—black), and 8000 (8000 IU—green) IU vitamin D3/kg chow IU/kg chow and serum 25-OH VitD was measured by ELISA monthly until 28 months of age (**A**). At baseline “B” and endpoint “E”, serum intact PTH was measured using ELISA (**B**), and bone density was measured using DEXA (**C**). “*” indicates *p* < 0.05 in 8000 IU compared to baseline, “***” *p* < 0.001 in 125 IU compared to baseline, “†” *p* < 0.05 in 8000 IU versus 1000 IU, “††” *p* < 0.001 in 125 IU versus 1000 IU, and “*ns*” denotes not significant.

**Figure 2 nutrients-12-03005-f002:**
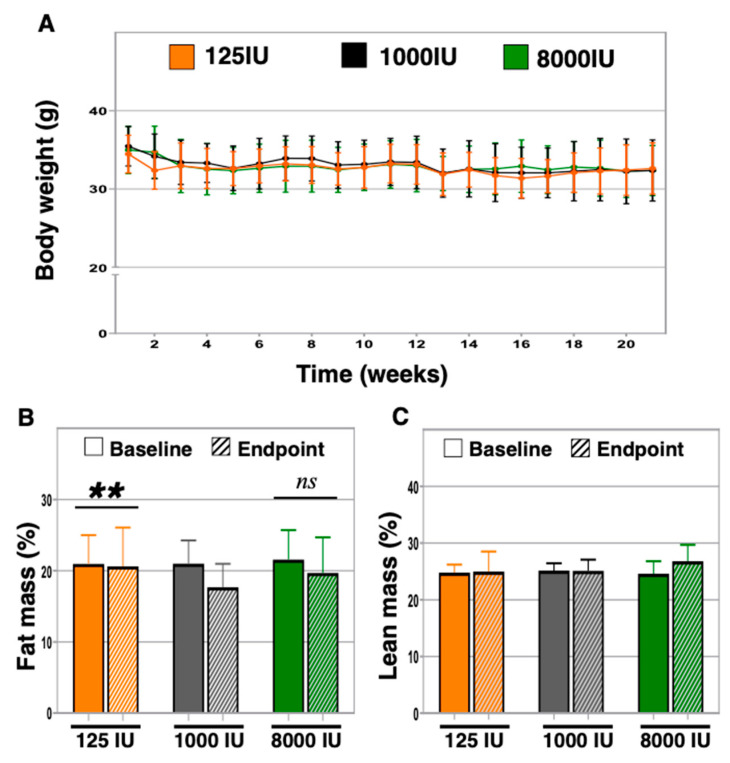
Analysis of body weight and body composition. Body weight was measured every two weeks in 24-month-old mice given varied amounts of vitamin D3 in chow (**A**). At baseline and endpoint, body lean mass (**B**) and fat percentage (**C**) were determined using DEXA. “**” indicates, *p <* 0.01.

**Figure 3 nutrients-12-03005-f003:**
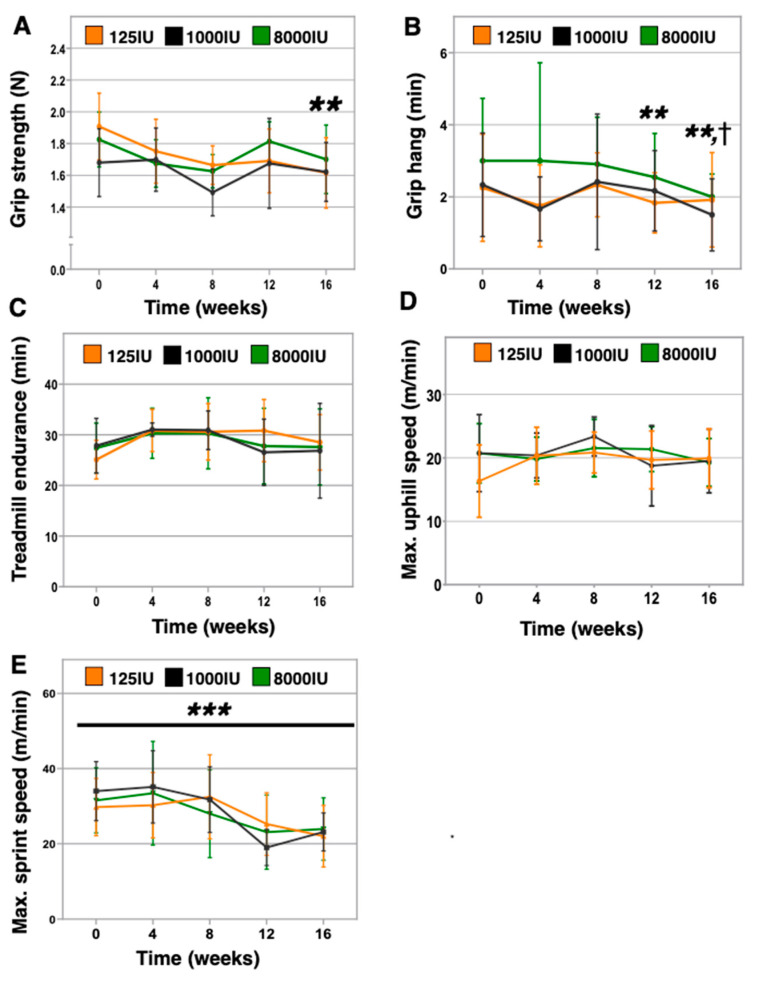
Physical performance in vitamin D insufficient, sufficient, and hypersufficient aged mice. A range of physical performance assessments were administered monthly in 24-month-old aged mice given varied levels of vitamin D3 in chow until 28 months of age. Assessments included grip strength as the best of 3 of 5 trials whereby mice are pulled away from a grid attached to a force meter until loss of grip (**A**); grip endurance in the best of two trials, in which mice are inverted on a grid and timed until loss of grip (**B**); treadmill endurance as a single trial to exhaustion whereby the treadmill belt increases from 5 to 35 m/min over 60 min (**C**); uphill treadmill intervals endurance as a single trial to exhaustion on an elevated treadmill (25°) with intervals that increase 1 m/min stepwise (**D**); flat treadmill sprint endurance as a single trial to exhaustion by which treadmill belt speeds increment 3 m/min stepwise (**E**). “†” *p* < 0.05 in 1000 IU versus baseline, “**” *p* < 0.01 in 125 IU versus baseline, and “***” *p* < 0.001 for all groups versus baseline.

**Figure 4 nutrients-12-03005-f004:**
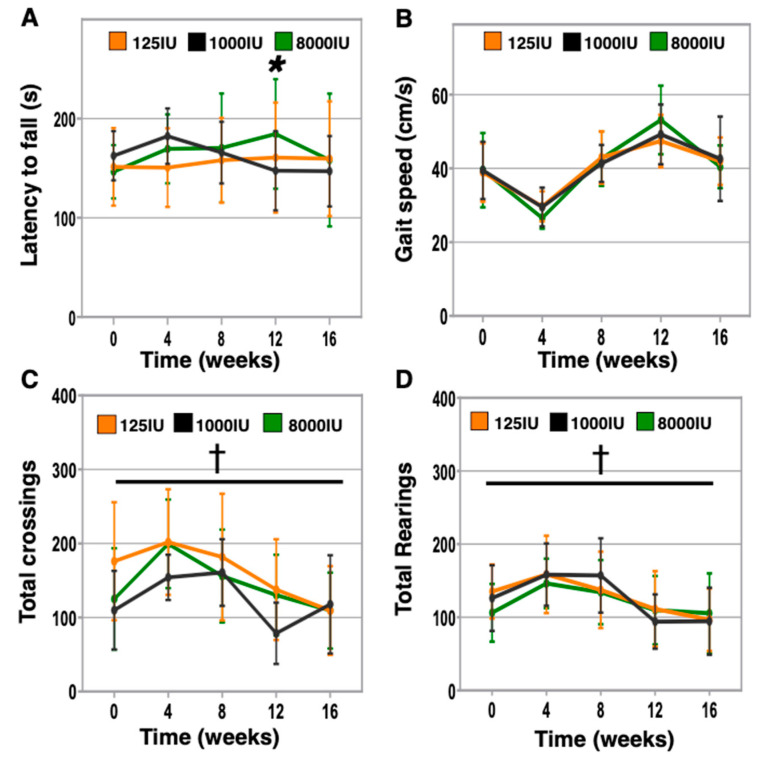
Vitamin D impacts on balance, gait speed, and activity levels. Additional physical performance characteristics were examined monthly in aged mice receiving varied vitamin D supplementation including: balance and coordination as the average time of the 2 best of 3 trials on a rotarod device that accelerates from 4 to 40 RPM over 5 min (**A**), gait speed as the average of the best 2 of 3 times to travel 1 meter (**B**), and activity as the total number of quadrant crossings (**C**) and rearings (**D**) in an open field arena in 30 min. “*” indicates *p* < 0.05 for 8000 IU versus baseline and “†” *p* < 0.001 for all groups versus baseline.

**Figure 5 nutrients-12-03005-f005:**
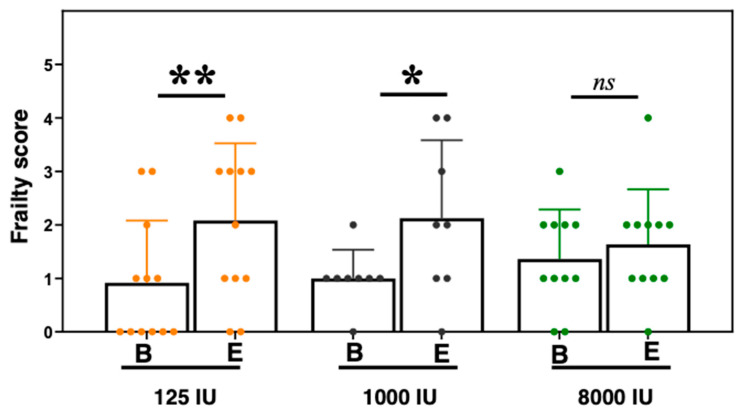
Frailty assessment in vitamin D treated mice. Frailty assessment for 125IU (orange), 1000IU (black), and 8000IU (green) mice is based upon the Fried et al. physical frailty scale [17], in which mice exhibiting three or more assessments below cohort-determined cutoff levels for unexpected weight loss, grip strength, endurance, activity levels, and gait speed were considered frail, and prefrail if below cutoff in 1 or 2 parameters. Scores were determined at baseline, “B” and endpoint “E”. Statistical significance is indicated by “*” *p* < 0.05, “**” *p* < 0.01, and *“ns”* indicating non-significance.

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
