# Peer review of "Vitamin D Insufficiency Reduces Grip Strength, Grip Endurance and Increases Frailty in Aged C57Bl/6J Mice"

_nutrients, 2020, doi:10.3390/nu12103005_

Round 1

Reviewer 1 Report

This is an interesting and important study of vitamin D in aging mice.

The authors have previously reported n dose effects of vitamin D on physical movement but this is the first study of aged mice and relates very well to the human condition of frailty.

Comments

  1. In the Abstract indicate doses of vitamin D are per kg chow.
  2. Discussion page 10: this sentence is repeated: “However, to our surprise in this study we observed few differences between vitamin D treatment groups for these assessments and others”

Author Response

We appreciate the positive feedback and constructive criticism for our manuscript. Our responses to reviewer #1 concerns are detailed below:

In the Abstract indicate doses of vitamin D are per kg chow.

Thank you for this feedback, we have updated the abstract accordingly

Discussion page 10: this sentence is repeated: “However, to our surprise in this study we observed few differences between vitamin D treatment groups for these assessments and others”

The extra line has been deleted.

Reviewer 2 Report

The research is very well conducted, but there are some problems that need to be addressed.

  • you only use male mice, why not look at both male and female. If you argue that mouse models are applicable to humans, you should also investigate the influence of sex on vitamin D levels. Generally, human females are considered to be more susceptible to vitamin D insufficiency/deficiency.
  • in the introduction, you stress the need for evaluating the relationships between frailty and serum 25-OH vitamin D, but in the end of the discussion you mention several studies that investigated the same in human groups. You did not make clear what your study added to the existing body of studies. How was your study different?

Minor comments:

Introduction:

  • The first sentence of your introduction is very vague. 
  • Can you really speak of geriatric populations? 
  • Why are geriatric individuals less exposed to sunlight and can you actually say their dietary intake is insufficient?
  • You did not mention that the human skin looses the capability to synthesize vitamin D with age, you should add this.
  • End of the introduction: although the relationship between frailty and vitamin D levels has been studied, you state it has not been studied in mouse models which would allow for better control of lifestyle and genetic factors. How would mouse models allow for a better control of human lifestyle? Human lifestyles are far more diverse than mouse lifestyles and human genetics may be more complex. You cannot say it like this.

Discussion:

  • you state you are the first to investigate the impacts on geriatric aged mice. The fact that it is the first time is not always relevant and is not synonymous to significant.
  • You state that there actually have been studies investigating the relationship between frailty and vitamin D levels, yet you did not clarify how your study contributes to our understanding. 
  • There is no discussing on how your mouse model aids in understanding the mechanisms, allows for a better control of human lifestyles, and how people will be allowed to age healthier.  Clarify this.

Author Response

We appreciate the positive feedback and constructive criticism for our manuscript. Our responses to reviewer #2 concerns are detailed below:

You only use male mice; why not look at both male and female. If you argue that mouse models are applicable to humans, you should also investigate the influence of sex on vitamin D levels. Generally, human females are considered to be more susceptible to vitamin D insufficiency / deficiency.

We feel this study introduces important impacts of serum vitamin D in the context of aged male mice that would be of interest to the readers of this journal. However, we agree about the importance of elucidating sex effects as it pertains to serum vitamin D levels and the subsequent impact on older females. We now add to the final paragraph of the discussion the following, “We also note the mice in our study were all male and that potential sex effects should be considered when extrapolating findings from this study to females. This is important given the findings of Burt et al that high doses of vitamin D supplementation were accompanied by progressive losses in total volumetric BMD that were greater in females than in males [46]. Further research will also be needed to parse out scenarios that are more clinically relevant such as when humans or mice are vitamin D insufficient for significant periods of time leading up to old age, particularly in female populations.”

Introduction: The first sentence of your introduction is very vague.

The first and now second sentences have been updated to read, “Frailty is a condition of greater susceptibility to adverse outcomes, including disability, loss of independence, and mortality. Additionally, frailty increases progressively with age, existing in 10% of individuals 65 and over and as much as 50% of those over the age of 85.”

Can you really speak of geriatric populations?

To avoid potential confusion with readers, we have replaced mentions of “Geriatric” populations as needed with terms such as aged mice, older adults, or otherwise specify specific ages.

Why are geriatric individuals less exposed to sunlight and can you actually say their dietary intake is insufficient? … You did not mention that the human skin loses the capability to synthesize vitamin D with age, you should add this.

Older individuals may exhibit reduced outdoor activity, particularly during disability, as well as reduced caloric intake, although we concur with the availability of supplements and recent trends are unclear whether older adults have less intake of vitamin D – and we have removed this notion from our introduction. The first paragraph of the introduction now reads, “Older adult populations are of particular concern for low serum vitamin D due to decreased sun exposure and a natural age associated decline of the generation of cholecalciferol (vitamin D3) from sunlight in skin tissues [5, 6].”

End of the introduction: although the relationship between frailty and vitamin D levels has been studied, you state it has not been studied in mouse models which would allow for better control of lifestyle and genetic factors. How would mouse models allow for a better control of human lifestyle? Human lifestyles are far more diverse than mouse lifestyles and human genetics may be more complex. You cannot say it like this.

We have clarified this statement. In short, mice are genetically identical and are housed in similar conditions and enrichments, receive the same chow compositions, and receive similar dark/light cycles – which in this case also includes reduced exposure to ambient UV-B. These controlled conditions thus allow a greater focus on the physiologic impacts of serum vitamin D levels on frailty status, which in humans is otherwise confounded by the great diversity of genetic and lifestyle contributions. This line therefore now reads, “Yet, the relationships between frailty and vitamin D have heretofore not been studied in a mouse model that can reduce genetic and lifestyle contributions via the use of genetically identical mice and maintaining mice with similar housing, lighting, and chow compositions.”

You state you are the first to investigate the impacts on geriatric aged mice. The fact that it is the first time is not always relevant and is not synonymous to significant.

This mention has been removed, the sentence now reads, “Here we investigate the impacts of differential vitamin D supplementation in aged mice.”

You state that there actually have been studies investigating the relationship between frailty and vitamin D levels, yet you did not clarify how your study contributes to our understanding … There is no discussing on how your mouse model aids in understanding the mechanisms, allows for a better control of human lifestyles, and how people will be allowed to age healthier.  Clarify this.
As suggested by the reviewer, we have added clarifying content to the final paragraph in our discussion of frailty. Specifically, we now write, “Here, we use a longitudinal model that allows observation of previously vitamin D sufficient aged mice that are then made insufficient for 4 months. In this context, only serum levels are changed while lifestyle and genetic factors are tightly controlled. This in turn allows a more focused examination of the physiological and functional impacts of serum vitamin D that are otherwise difficult to accomplish in a human clinical trial. Here, we identified that while the 1000IU and the 125IU mice exhibited a statistically significant increase in frailty (Figure 5), the 8000IU mice did not. These data suggest that higher than recommended supplementation and/or sun exposure may be necessary to attenuate frailty progression in older adults, which would require longer term human clinical trials (>5 years) to validate.”

In the introduction, you stress the need for evaluating the relationships between frailty and serum 25-OH vitamin D, but in the end of the discussion you mention several studies that investigated the same in human groups. You did not make clear what your study added to the existing body of studies. How was your study different?

The novel aspects of our study involve modulating and then maintaining serum vitamin D levels in an aged organism at levels considered insufficient, sufficient, and hyper-sufficient for a human. Further, we apply frailty assessment in a longitudinal manner across 4 months, which arguably would be equivalent to a 5-10 year period in a human. No such human study has been accomplished, in part due to the challenges of conducting a long term clinical trial as well as ethical considerations of making a human vitamin D insufficient for this period of time. We have hopefully captured and expressed to the readers these novel aspects of our study.

Reviewer 3 Report

it’s nice article. Though i would appreciate if the author can incorporate some xray images of those mice that will clearly show and confirm the data. 

Author Response

We appreciate the positive feedback and constructive criticism for our manuscript. Our responses to reviewer #3 concerns are detailed below:

It’s nice article. Though i would appreciate if the author can incorporate some xray images of those mice that will clearly show and confirm the data.
As suggested we now include a new supplementary figure that shows representative X-ray images of mice closest to the group mean for bone mineral density respective to each group.

Round 2

Reviewer 2 Report

All my suggestions have been dealt with in a satisfactory way and I look forward to the publication.